# Investigation of Severe Plastic Deformation Effects on Magnesium RZ5 Alloy Sheets Using a Modified Multi-Pass Equal Channel Angular Pressing (ECAP) Technique

**DOI:** 10.3390/ma16145158

**Published:** 2023-07-21

**Authors:** Partha Sarathi Sahoo, Manas Mohan Mahapatra, Pandu Ranga Vundavilli, Rama Krushna Sabat, Sachin Sirohi, Sanjeev Kumar

**Affiliations:** 1School of Mechanical Sciences, Indian Institute of Technology Bhubaneswar, Kansapada 752050, India; pss11@iitbbs.ac.in (P.S.S.);; 2School of Minerals, Metallurgical and Materials Engineering, Indian Institute of Technology Bhubaneswar, Kansapada 752050, India; 3Department of Mechanical Engineering, Delhi NCR Campus, SRM Institute of Science and Technology, Modinagar 201204, India

**Keywords:** Mg-RZ5, ECAP, UFG, channel angle, severe plastic deformation

## Abstract

The present study investigates the effects of multiple passes of equal channel angular pressing (ECAP) on magnesium alloy sheets with the assistance of an Inconel plunger along with a die setup having a channel angle of 120° and corner angle of 10° operating at a temperature of 200 °C followed by the required heat treatment processes. The microstructural analysis of the sheet samples at various stages of the multi-pass hot ECAP has shown evidence of ultrafine grain refinement (UFG) due to the occurrence of severe plastic deformation. X-ray diffraction analysis has also exhibited the presence of phases like MgZn and CeZn_3_ which is supposedly responsible for the enhancement of the mechanical properties. As a result, the room temperature tensile and compressive strengths have improved by 6.12% and 6.63%, respectively, after the second pass, and 11.56% and 15.64%, respectively, after the fourth pass of ECAP. Additionally, the hardness of the sheets has increased by 6.49% and 16.64% after the second and fourth pass of hot ECAP, respectively, mainly attributed to the drastic decrease in grain size from 164 μm to 12 μm within four ECAP passes, all these with a negligible change in ductility. This success in the thermomechanical processing of Mg-RZ5 alloy sheets using a die channel angle of 120° with a minimal number of passes of hot ECAP under a controlled equivalent strain, further opens doors for incorporating optimizations and/or additional aspects so as to achieve even better grain refinements, and consequently, mechanical strength improvements thereby catering to the industrial needs of aerospace and construction areas.

## 1. Background and Introduction

The excellent strength-to-weight ratio of magnesium and its alloys makes them highly desirable among the lighter metals. However, due to their poor formability at lower temperatures and the ensuing problems frequently experienced in component shaping procedures, magnesium alloys have only seen a limited amount of use in industrial applications. This restriction has a significant side effect in that the manufacturing and fabrication of magnesium alloys often take place at moderate-to-high temperatures. Magnesium alloys undergo thermo-mechanical processing, which results in changes to the material structure, notably to the grain size and its distribution/uniformity. In contrast to the wrought materials produced by hot extrusion or rolling, which are often characterized by an inconsistent distribution of fine and coarse grains, standard cast structures are distinguished by coarser grains of several hundred microns [1]. The development of fine grains by dynamic recrystallization at the grain boundaries during its processing at controlled temperatures is the direct cause of the multi-modal distribution of grain sizes as observed in wrought magnesium [2]. However, in practical terms, the associated reduction in the cross-sectional areas of the samples substantially restricts the overall deformation that can be obtained by traditional thermo-mechanical processing. This form of processing thus frequently does not promote a uniform refinement of the original coarser structure in the material and rather results in a multi-modal grain size distribution.

The limitation on the extent of deformation that may be imparted on a material during processing has been eliminated, largely owing to the experimental advancement of processing techniques in recent years involving the utilization of severe plastic deformation (SPD) like equal-channel angular pressing (ECAP) [3], high-pressure torsion [4], multi-directional forging [5], and cryorolling [6]. Presently, in an ECAP technique, a material sample having a cross-sectional shape identical to that of the channels undergoes deformation via extreme pressure as it is passed through a die featuring two or more channels of equal cross-sectional shapes that intersect at a channel angle denoted by ϕ, and an outer corner angle represented by Ψ [7]. Given that the cross-sectional area of the workpiece and overall dimensions remain unchanged following to the application of the process, it may be subjected to repeated the equal channel angular pressing (ECAP) procedure in the form of routes A, B_A_, B_C_, and C, with the aim of inducing significant strains. Every instance of deformation applied during this procedure is referred to as a pass [8]. The equivalent strain value can be empirically evaluated [9] using Equation (1) as below, where N denotes the number of passes.
(1)$$\varepsilon =\frac{N}{\surd 3}\left[2\;cot\left(\frac{\Phi }{2}+\frac{\Psi }{2}\right)+\Psi \;cosec\left(\frac{\Phi }{2}+\frac{\Psi }{2}\right)\right]$$

During experimental practices, a billet can be rotated about its longitudinal axis to introduce various slippage systems [10,11] with each pass which has led to the identification of various basic fabricating routes as follows: Sample is not rotated in route A;The sample undergoes rotation either in the opposite directions of 90° along route B_A_ or in a consistent direction along route B_C_;180° overturning rotations of the sample through route C [12].

These state-of-the-art techniques can be used to enhance the mechanical properties of low-ductility and high-utility alloys like the magnesium and titanium alloys [13], most notably those having rare earth alloying elements. Various extrusion ratios have been tested on Mg-AZ31 alloy, so as to demonstrate its machinability properties in terms of milling and drilling operations [14]. To produce materials with evenly distributed arrays of ultrafine grains and better mechanical characteristics, including extraordinary superplastic capabilities, SPD methods were therefore introduced for the manufacturing of magnesium alloys. The approach to grain refinement in magnesium alloys treated by ECAP differs significantly from the mechanism witnessed with f.c.c. metals like aluminium and copper, which became evident in practice during this time [15]. The initial passes of ECAP result in an array of elongated subgrains in the f.c.c. metals, with the grain boundaries having low angles of misorientation and the longer axes aligned parallel to the principal slip system [16,17]. After approximately four passes of ECAP, additional passes typically develop additional dislocations, which then re-arrange and disintegrate in accordance with the theory of low-energy dislocation structures [18] to produce a variety of fairly equiaxed grains divided by boundaries with large angles of misorientation. High-purity aluminium is one f.c.c. material where the development of this type of grain structure has been extensively studied [19]. High-performance processes, like a novel grinding method, have also shown the effects of high strains and low pressure on the Inconel 718 superalloy in the form of better lifespan and surface finish [20,21]. Conversely, as the number of ECAP passes increases, the original larger grains in magnesium and its alloys are eventually consumed by the new, smaller grains which develop along the original boundaries during the SPD process [22]. The improvement in mechanical properties can provide extra lifespan for the magnesium alloys being used in construction fields and aerospace industries as thin sheet layers to the bodies [23]. These are very useful in the fabrication of biomedical implants and prosthetics, as well as contributing to the manufacture of aircraft engines, gearbox casings, and housings in high-performance bikes and other sporting equipment [24]. 

Earlier simple models have described grain refinement in magnesium alloys that provided a general understanding aimed to determine the forecasting efficacy of predicting the novel grain structures that might develop in magnesium alloys when subjected to various working conditions [25]. Various ECAP processing settings and the microstructure of the material before undergoing ECAP have an impact on the aspects of this grain refinement mechanism in magnesium alloys [26]. Diverse microstructural characteristics, such as variable final grain dimensions and distinct multi-modal dispersions of grain sizes, have been observed in the case of such materials following the ECAP processing by virtue of the multiple processing and structural permutations that may be conceivable [27]. However, the route C of the ECAP technique has been comparatively less explored, and moreover, alloys in the form of sheets have also been less studied. Most of the research works have been for different alloys and pure metals conducted in the form of cylindrical billets. A sheet metal study can provide a comprehensive idea from a new perspective, and further cater to the needs of aerospace and automobile industries as body frames and metallic covers. The goal of the current study is to investigate the mechanical and microstructural characteristics pertaining to an RZ5 magnesium alloy processed by multiple hot ECAP passes and to create a systematic method for ultrafine grain refining that could be applied to the processing of other magnesium-based alloys with a similar composition. It is crucial to underline that the present report elucidates the intricate mechanism of grain refinement required to allow for the development of various structures depending on the processing factors and, additionally, it directly affects the most effective processing routes for the efficient fabrication of such materials with lower ductility. The current study also reveals a conducive approach in terms of the forming temperatures and number of passes, to produce good grain refinement and material properties, so as to effectively fulfil the purpose.

## 2. Materials and Methods

For conducting the present work, the Mg-RZ5 alloy was procured in the form of ingots from Hindustan Aeronautics Limited, Bangalore, India. Furthermore, the ingots are machined and cut into sheets of 3 mm thickness precisely with the help of wire-cut electrical discharge machine. The elemental composition of the cast alloy is depicted in Table 1.

This composition of the alloy as-received has also been verified with the help of energy dispersive x-ray spectroscopy using a ZEISS MERLIN Compact field emission scanning electron microscope (FESEM) of the India make. The mechanical and microstructural properties of the magnesium alloy were investigated after the second- and fourth-pass subjected to equal channel angular pressing. The schematic diagram of a general ECAP technique is shown in Figure 1. 

To produce uniform shear throughout the bulk of the material, different routes (i.e., Routes A, B_A_, B_C_, C [29,30]) are used during an ECAP process, but to maintain the uniformity of shear deformation throughout the body in the form of sheet metals, the present experimental study was based on the route C of ECAP. The idea and processing of route C was provided in the form of Figure 2. Being a sheet alloy with a rectangular cross-section, the available route alternatives are Route A and Route C. The 90° rotations to be imposed in Routes B_A_ and B_C_ are not possible using a fixed channel path for the samples that are not cylindrical in shape. Furthermore, Route A does not bring uniformity in grain refinement as it imposes shear strains mostly on a single surface of the alloy, and therefore is the least effective among all the available routes [31]. Thus, Route C is understood to be the best alternative in case of a ECAP pass for the samples with square and rectangular cross-sections. The adopted Route C recurringly and completely overturns the alloy sheet by 180° on its vertical axis, after undergoing a single pass of ECAP and fully exiting the channel.

### 2.1. Die Designs at the Pre-ECAP Stage

The pre-processing stages comprise the designing of a compact and sustainable setup so as to accomplish the multiple passes of hot ECAP. It is to be noted that the magnesium alloy sheet will be imposed with a cumulative strain value of ~1.1 at each pass while the alloy is being pre-heated to a temperature of 200 °C. Thus, the die units manufactured from H13 steel and the plunger fabricated from inconel 617 alloy were designed accordingly to accumulate the magnesium alloy in the designated slot on the plunger surface, thereby deforming the alloy sheet at the stopper point of the die. Figure 3a shows the top, front, sectional, and isometric views of the bigger unit of the die setup which is approximately three-fourths of the whole setup. Similarly, Figure 3b displays the different views of the smaller unit of the die setup which houses the stopper part as well as the exit channel path for the deformed magnesium alloy sheet. It can be seen that the bigger and smaller units are cut in such a way that they can accommodate the channel angle of 120° at their junction. Their assembly is performed using a sliding press fit and having additional support from four dowel pins in the form of M10 bolts, as can be seen in Figure 3d. Furthermore, the cyclindrical plunger of 30 mm in diameter is machined to have a flat surface to contain the magnesium alloy sheet in the fabricated slot, with the dimensions shown in Figure 3c. This plunger is then pressed from the top of the die setup with the help of the drawing pressure from a hydraulic press machine. Clearly, these CAD drawings were articulated prior to the machining processes to manufacture the complete die assembly.

### 2.2. Complete Experimental Setup for ECAP Processing

In this experiment, a cylindrical plunger with a 30 mm diameter manufactured of Inconel-617 alloy is employed as support to counteract the effects of buckling. The sheet of Mg-RZ5 alloy has dimensions of 105 × 10 × 3 mm. The cylindrical plunger is designed to have a flat surface, as shown in Figure 4a, so that a slot with dimensions identical to those of the magnesium alloy sheets may be produced on its front face. The smaller and larger pieces of the constructed die arrangement, which are made up of H13 die steel, are pushed into the designated through-hole from the top side, as seen in Figure 4b,c, respectively. The exit point from the 120° channel path of ECAP enabling the flow of the alloy sheet material is included in the smaller unit. Additionally, it has a stopper with dimensions identical to those of the sheet—3 mm in length and 10 mm in width, as can be seen in Figure 4b, which prevents it moving downwards and directs it at an angle of 120° along the extrusion path of the ECAP channel. According to Figure 4c, the larger die unit incorporates a sliding press-fit feature that allows it to be assembled with the smaller unit via M10 bolts. The total size of the die setup unit measures 200 × 160 × 75 mm and has a through-hole for the passage of the plunger. Additionally, as shown in Figure 4d, this assembly is housed inside a heating setup that employs the conductive method of heat transfer and has a total capacity of 5 kW power connected to an auto-variac transformer capable of producing temperatures greater than 800 °C. To carry out the ECAP experiments, a dual-acting double-platen hydraulic press with a total capacity of 100 tonnes, as illustrated in Figure 4e, is employed for pressing the plunger with the alloy sheet in its allotted slot through the die hole. In the current investigation, experiments were carried out using a drawing pressure of 20 tonnes at a temperature of 200 °C after accounting for a safety factor. All the samples have been subjected to a short annealing time of 10 min at a temperature of 100 °C.

### 2.3. Material Characterization

Figure 5 shows the magnesium alloy sheets before and after the SPD processing, where Figure 5a depicts the alloy in its base form before ECAP, and Figure 5b,c depict the alloy samples after being subjected to two and four passes of hot ECAP technique, respectively. It can be observed that there has been no major changes in the dimensions of the sheet sample before and after the ECAP technique. The severe plastic deformation processes are different hybrids of metal forming techniques, which does not involve material removal, and mostly concentrate on improving the material properties by imposing strains. However, there has been a noticeable distortion failure at the bottom edge of the sample after multiple passes of ECAP, as the edge portion sustains very high strains, being the point of impact of the drawing force when the sample starts to bend at the channel angle. The initial and final samples were prepared for microstructural analysis with stepwise polishing and further chemically etched using the acetic picral solution. The chemical composition for 95 mL of this etchant has a specific ratio of 70 mL of ethanol, 10 mL of distilled water, 11 mL of acetic acid, and 4 mL (4.2 g) of picric acid. Then, the grain structures of the magnesium alloy, after multi-pass deformation due to ECAP, were examined using a ZEISS MERLIN compact field emission scanning electron microscope (FESEM, Bengaluru, India). For the precise microhardness evaluation of the samples, the Omnitech-manufactured semi-automatic Vickers Hardness Tester (Rajkot, India) was utilised with a load of 200 gf and a dwell time of 10 s. To study the tensile properties, the samples are subjected to the tensile loading of 20 kN at a slow strain rate of 0.5 mm/min with the help of a Shimadzu made ultimate testing machine (Kyoto, Japan) having a total load capacity of 50 kN. The compressive stress–strain properties were acquired using the universal testing machine of BISS make (Bengaluru, India) with a loading capacity of 100 kN. The phase study of the Mg-RZ5 alloy was accomplished with the help of the BRUKER D8 Advance X-ray diffractometer (Billerica, MA, USA) with Cu K_α_ radiation (λ = 1.542 Å) over a scanning range of 20°–80° and a scan speed of 0.5°/min. To ensure the correctness of the sample data, several readings and pictures of the samples were taken with varying magnifications wherever necessary.

## 3. Results and Discussion

Different approaches exist for the enhancement of the mechanical characteristics of metal alloys based on the severe plastic deformation techniques conducted for regulating the dislocation density, grain refinement, and size of the various phases. The impact of these aspects on the mechanical properties of magnesium RZ5 alloy sheets is the matter of study for the following segments using the current die setup and sheet sample dimensions. In the present research work, the alloy sheets are being deformed at a uniform ramming speed of 5 mm/s to incorporate the von Mises equivalent strain of ~1.1 per pass for a maximum of four consecutive passes of hot ECAP, and the effects of the hot ECAP are being further examined in terms of their mechanical and microstructural properties.

### 3.1. Effects of ECAP on Room-Temperature Tensile and Compressive Properties

The relation between engineering stress and engineering strain has been plotted from the outputs of the tensile tests of Mg-RZ5 alloy sheets conducted before and after ECAP, as shown in Figure 6. It can be observed that the ultimate tensile strength just before the necking failure has increased for the post-ECAP samples. Though the yield strength of the post-ECAP samples shown lesser improvements compared to that in the base alloy, these magnesium alloy sheets after two and four passes of hot ECAP witnessed a major increase in stress-bearing capacities in terms of their ultimate tensile stresses. Similarly, the relationship between the compressive stress and compressive strain was traced for the Mg-RZ5 sheets (devoid of the buckling effects during testing) under various conditions of the severe plastic deformation conditions as opposed to its base form, as shown in Figure 7. It can be seen that the magnesium alloy samples after two and four passes of hot ECAP have shown excellent responses towards the compressive stresses and buckling loads, as compared to that in the base alloy before ECAP. Thus, the ultimate compressive strengths saw a major rise for the post-SPD samples as opposed to that in the pre-SPD samples, before dipping towards their failure zones under crushing and compressive forces. Furthermore, Figure 8a shows that the ultimate tensile strength of the sample after ECAP improved by 6.12% and 11.56% compared to that of the base alloy, after the second and fourth pass of hot ECAP, respectively. These features are potentially useful in high-altitude and high-pressure zones, for example, an aircraft body. 

Furthermore, to understand how a material would behave under crushing stresses, the compression tests are conducted which can typically measure the elastic and compressive fracture properties of the brittle or low-ductility metals and alloys. Figure 8b shows that the compressive strengths of the magnesium alloy improved by 6.63% and 15.64%, after being subjected to two and four passes of ECAP, respectively, with the help of route C. As can be observed in Figure 8a and Figure 8b, the ultimate tensile and compressive strengths of the Mg-RZ5 alloy sheets increased from 147 MPa and 202.6 MPa to 156 MPa and 216 MPa, respectively, after the second pass of hot ECAP, and to 164 MPa and 234 MPa, respectively, after the fourth pass of the hot ECAP technique. 

The UFG alloy produced via non-isothermal ECAP exhibits higher strength properties and low plasticity supposedly due to the high dislocation density, ultrafine grain refinement and better grain size distribution. As compared to as-received Mg-RZ5 base alloy, both strengths at ambient temperatures improved with a negligible compromise in its ductility along with the utilization of a multi-pass hot ECAP technique and following annealing treatments due to its fine and uniform h.c.p. microstructure.

### 3.2. Evaluation of Embrittlement Induced in the Alloy Sheet Due to Multiple Passes of ECAP

There has an induction of brittleness in the magnesium alloy sheet samples due to the multiple passes of hot ECAP where the hardness has increased from 60.1 VHN in its base condition to 64 VHN and 70.1 VHN, recorded after the second and fourth pass, respectively, as can be seen in Figure 9. The reason for this phenomenon may be due to presence of phases like MgZn and CeZn_3_ occurring due to dynamic recrystallization caused from the hot deformation processes. This means that the extreme metal forming technique has brought about an improvement of 6.49% and 16.64% in its hardness value after two and four passes of the sheet alloy, respectively.

Researchers [33] previously indicated that the dislocations become annihilated by the sub-grain boundaries, whereas the new recrystallized grains emerge along the high-angle grain boundaries during dynamic recrystallization. The average grain sizes including their standard deviations are evaluated after the second and fourth pass of the ECAP technique from the PANalytical X’Pert Highscore Plus software (2019 version) based on the peaks analyses from the monitoring of the x-ray diffraction data. It can be noticed from Table 2 that the mean value of the grain size, as seen in the base alloy, has substantially decreased due to imposed strains from the subsequent passes of the hot ECAP technique.

Thus, due to this reduction in grain size in the microstructure of the magnesium alloy owing to the severe plastic deformation technique utilized for the present study, it can be concluded that there has been an occurrence of ultrafine grain refinement thereby bringing about the enhancement in the mechanical properties in the alloy. As already mentioned in the preceding section, the improvement in the hardness and tensile strength values of the Mg-RZ5 alloy clearly reflects this reduction in grain size. The Hall–Petch relationship [34], which shows that for most metals and alloys, the stress and hardness will rise with a reduction in average grain size, can also be used to support the inference: (2)$$\sigma =\sigma_{o}+\frac{K_{y}}{\surd d}$$
(3)$$H_{V}=H_{o}+\frac{K}{\surd d}$$
where σ_o_ is the lattice frictional stress, K_y_ represents yielding constant, and H_o_ and K are material constants.

### 3.3. Grain Refinement Exhibited Due to SPD

Figure 10 illustrates the FESEM micrographs of the magnesium alloy sheets in its base form as shown in Figure 10a,b, followed by the images of the alloy sheets being subjected to various stages of severe plastic deformation (i.e., different number of passes of hot ECAP) as shown in Figure 10c,d (for two passes), and Figure 10e,f (for four passes). Images were procured from the sheet cross-sections in the direction of extrusion for both the conditions of deformation. It can be observed in Figure 10d that the grain boundaries are being more prominent which hints to the fact that the subgrains are actively in the process of appearing along and around the hexagonal grain boundaries in the microstructure of magnesium alloy. This phenomenon can significantly develop finer grains at the surroundings of the hexagonal boundaries of the coarser grains. Furthermore, the evidence of blisters as shown in Figure 10f can be attributed to the effects of high forming temperatures used for the ECAP technique after being imposed with a significant amount of shear stresses due to the multiple passes using route C.

Prior research has indicated the presence of dynamic recrystallization in magnesium alloys have undergone deformation within the temperature range of 150–350 °C [35]. This temperature range is frequently utilized for the assessment of magnesium alloys via ECAP [36,37]. The refinement of the internal cores of the coarse grains is absent because the new fine grains preferentially develop along twin boundaries in a necklace-like configuration and at the primary boundaries of the coarse-grained structure during dynamic recrystallization. An early theory for the dynamic recrystallization-induced grains nucleation at the grain boundaries proposed to be dependent upon the anisotropy pertaining to its shear stresses required for the activation of various slip systems associated with a hexagonally close-packed structure [38]. It is projected that the systems with a simple slip are supposed to get triggered at the grain cores, which will not be enough for the achievement of a proper grain refinement. In actuality, higher stresses are caused by the incompatibility in slip between the adjacent grains at grain boundaries. The more complicated slip systems are activated by these higher stresses, which also promote the growth of three-dimensional arrays of dislocations and the dynamic recrystallization in these regions [39]. Therefore, the number of locations accessible for the nucleation of newer grains determines how well the grains of the magnesium alloy may be refined [40].

As discussed earlier, the phenomenon of dynamic recrystallization gives rise to the formation of subgrains which results in grain refinement due to severe plastic deformation processes. Figure 11a indicates the initiation of coarse subgrains formation at the grain boundaries region of magnesium alloy after the second pass of hot ECAP, whereas Figure 11b gives evidence of the development of refined subgrains with their own grain boundaries after the fourth pass of the hot ECAP technique. Though there is the small presence of subgrains in the form of coarse residues after the fourth pass, there has been a clear segregation from the finer subgrains along with the formation of their grain boundaries with higher lattice energy. Thus, this provides a clear idea regarding the achievement of ultrafine grain refinement in the magnesium alloy due to the imposed shear stresses in the SPD process [41] in the form of modified multi-pass hot ECAP used for the current investigation. Also, the grain boundaries of these subgrains have stayed quite stable after the appropriate annealing treatment after the ECAP procedure. Therefore, the grain boundaries of the coarser grains as shown in Figure 10d can be considered as the secondary phase of the alloy microstructure.

In addition, the FESEM images of tensile fractography surfaces of the alloy sheet samples after different severe plastic deformation conditions are shown in Figure 12. Prior to that, the condition of the base alloy after a tensile testing is also depicted in Figure 12a,b with varying magnifications. Then, after being subjected to the second pass of the hot ECAP, the microstructure at the necking zone of its tensile tested condition has shown evidences of initiation of micro-cracks and the presence of tear ridges. Upon higher magnification, it has also shown evidence of the presence of dimples, which is an indication of the micro-voids being active in a cyclic mode that tend to appear during the plastic flow of a material and sometimes eventually coalesce into larger voids. This phenomenon implies that the ductility of the magnesium alloy sheet has been more or less unaffected after the ECAP processing. These dimples appear to be elongated in the direction of shear due to the immense shear stresses involved in the procedure. Additionally, Figure 12e,f show that the fourth pass of the hot ECAP technique has finally given rise to the creation of microvoids that were initiated after the second pass. Better magnification of these microvoids reveal the intergranular cleavages that tend to go around the individual metal grains. These intergranular cleavages are also sometimes called a “rock candy fracture” because of such appearances of the exposed grain structure in the material. This cleavage is an evidence of embrittlement due to the increase in the hardness of the material.

For the face-centred cubic (FCC) metals like Al and Cu, the ductility of the sheets after hot forming tends to improve rather than that of room temperature formed sheets [42]. However, for the hexagonal close packed (HCP) magnesium alloys, however, the effects of hot ECAP over the ductility of sheets appear to be insignificant, as evident from the current investigation.

### 3.4. Phase Analysis Post-ECAP

Dual-phase alloys and metals can also be noticed to have exceptionally high strength and ductility [43]. Li et al. [44] recently reported the development of a dual-phased alloy that eradicates the strength-ductility trade-off. Restricting and supervising the scale of the secondary phases in case of the alloys has become important for the overall improvement of the material properties. Figure 13 reveals the presence of phases like Mg_17_Ce, MgZn, and CeZn_3_ in the Mg-RZ5 alloy subjected to multiple passes of the hot ECAP technique. There was no phase formed between Mg and Zr because no chemical reaction is possible at this range of the working temperatures. The MgZn phase formation proves to be adding up to the hardness of the magnesium alloy, whereas the Mg_17_Ce and CeZn_3_ phases tend to improve the tensile and compressive strength properties at elevated temperatures of the working conditions. A few oxide peaks may be present during the phase analysis; however, apart from MgO, they could not be traced by the x-ray diffractometer due to their low intensities. This analysis was conducted using the JCPDS database.

### 3.5. Elemental Distribution at Various Stages of the Multi-Pass ECAP Technique

Figure 14 and Figure 15 show the dispersive spectroscopy images of the distribution of composition elements like magnesium (Mg), zinc (Zn), zirconium (Zr), cerium (Ce), and oxygen (O) after two and four passes of hot ECAP, respectively. It can be noted that the thickness of the Mg grain boundaries in Figure 15 after the fourth pass of hot ECAP is lesser and more evenly distributed than that in Figure 14 taken after the second pass of hot ECAP. It can also be seen that the distributions of Zr and Ce in the dispersion spectroscopy images have more or less stayed consistent under both the conditions of hot ECAP technique, i.e., after the two and four passes. O is the element responsible for the formation of oxides while working at higher temperatures involved during the metal forming techniques like the one in the present study. The zinc element, however, has seen a slightly higher presence in the dispersion spectroscopy studies after the fourth pass of the ECAP technique, than that after the second pass, though the distribution has remained almost same.

Similarly, point distribution spectral studies have also been undertaken at both the deformation conditions of the ECAP technique, in support of the dispersive spectroscopy studies for examining the elemental distribution. Figure 16 corresponds to the energy dispersive x-ray spectroscopy (EDS) analyses for both conditions, i.e., before and after ECAP. From the elemental analyses in Figure 16a,b, it is observed that the weight percentages of Mg, Zn, Zr, and Ce are 93.91, 4.14, 0.63, and 1.32% for base alloy, respectively, which is nearly equivalent to the standard elemental composition of Mg-RZ5 alloy. Following the analysis of the EDS spectra after two passes of hot ECAP, the weight percentage of magnesium had notably reduced to 93.33%, whereas those of Zn, Zr, and Ce improved to 4.47, 0.76, and 1.44%, respectively, as shown in Figure 16c,d, which might be due to the phenomenon of the initial stages of ultrafine grain refinement. Furthermore, upon increasing the number of passes while keeping the material forming temperature constant, it was found in the EDS spectra that the weight percentage of the main element, i.e., magnesium, again dipped to 92.78%, whereas the weight percentages of the other alloying elements including the rare-earth metal have increased steadily. As can be seen in Figure 16e,f, the weight percentages of Zn, Zr, and Ce progressed to 4.76, 0.91, and 1.55%, respectively, most probably due to the achievement of ultrafine refinement, i.e., the formation of subgrains and the segregation of finer grains around the coarser grains and their grain boundaries.

Additionally, working out the EDS spectra and associating them with the phase analyses as discussed in the previous section, it can be concluded that the changes in weight percentages of the alloying elements must be due to the dynamic recrystallization during the thermomechanical processing in the multiple passes of hot ECAP.

## 4. Summary and Conclusions

Route C of the ECAP technique was utilized to refine the microstructure of Mg-RZ5 alloy sheets in multiple passes. The microstructural evolution and room temperature mechanical properties of the Mg-RZ5 alloy post-ECAP were discussed and the following conclusions could be drawn from the present study:The multi-pass hot ECAP of a rare-earth metal-based magnesium alloy was successfully conducted with the preheating of the sheet specimens to 200 °C, similar to that utilized in the previous literature for various other magnesium alloys, with a maximum equivalent plastic strain of 1.2.The microstructure evolution has demonstrated the achievement of good grain refinement during the hot ECAP process with an increasing number of passes, with the formation of sub-micrograins. Dynamic recrystallization has occurred which resulted in a drastic decrease in grain size of the alloy from 164 μm in as-procured magnesium alloy to 51 μm and 12 μm after two and four passes of hot ECAP.The ultimate tensile and compressive strengths of the Mg-RZ5 alloy sheets increased from 147 MPa and 202.6 MPa to 156 MPa and 216 MPa, respectively, after the second pass of hot ECAP, and to 164 MPa and 234 MPa, respectively, after the fourth pass of the hot ECAP technique. As compared to as-received Mg-RZ5 base alloy, both strengths under ambient temperature conditions improved with a negligible compromise in its ductility with the utilization of the multi-pass hot ECAP technique and following annealing treatments due to the fine and uniform microstructure.The hardness increased from 60 VHN in its base condition to 64 VHN and 70.1 VHN recorded after the second and fourth passes of hot ECAP, respectively. The reason for this phenomenon is the development of phases like MgZn and CeZn_3_ occurring due to the dynamic recrystallization caused by the hot deformation processes.

## Figures and Tables

**Figure 1 materials-16-05158-f001:**
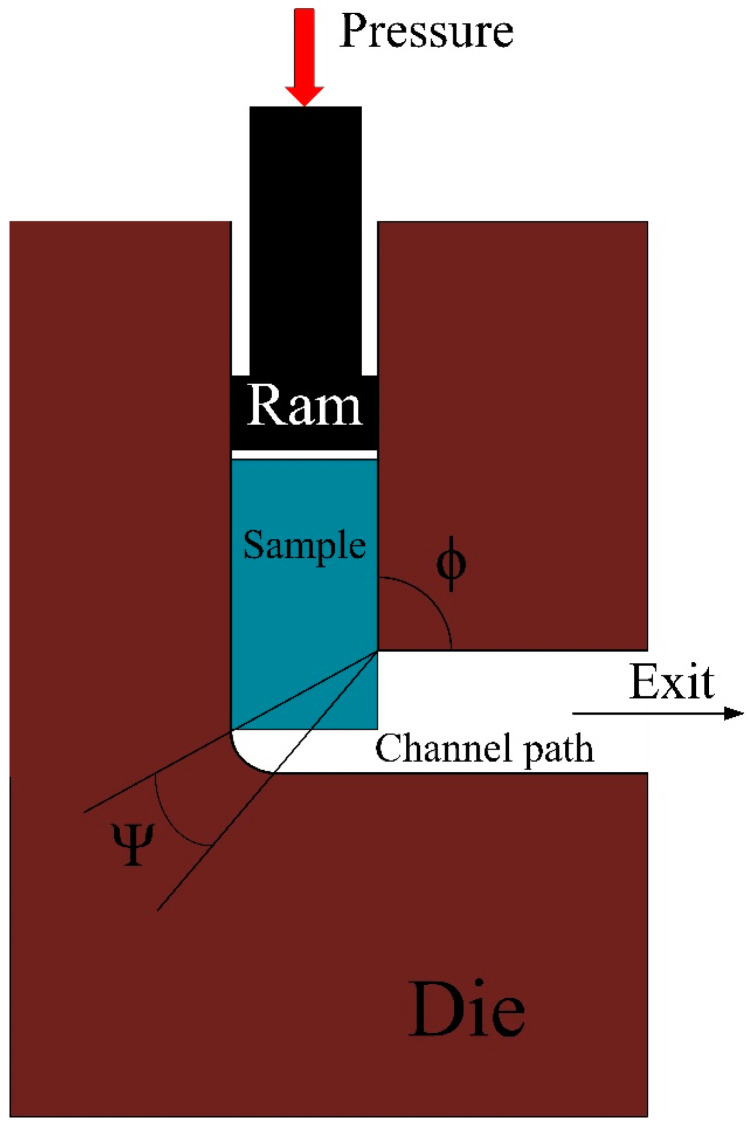
Schematic of a typical ECAP [28].

**Figure 2 materials-16-05158-f002:**
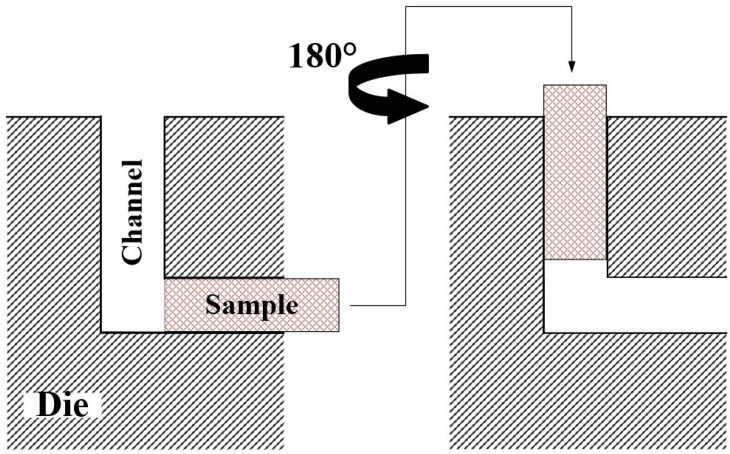
Adaptation of Route C for conducting the multiple passes of ECAP technique [32].

**Figure 3 materials-16-05158-f003:**
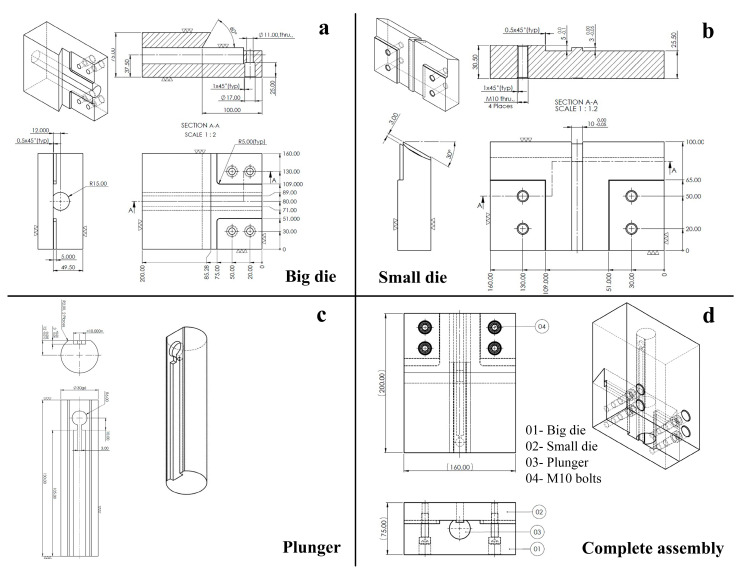
Drawings of top, front, sectional, and isometric views of (**a**) bigger unit (**b**) smaller unit (**c**) plunger (**d**) complete assembly, to design the ECAP die setup.

**Figure 4 materials-16-05158-f004:**
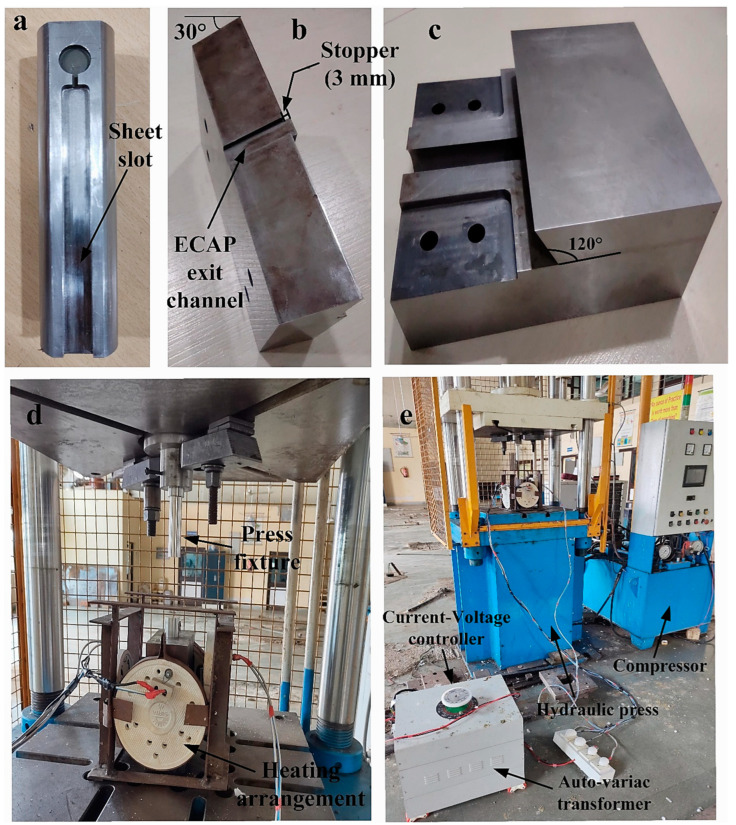
(**a**) Plunger with slot for Mg-RZ5 sheet; (**b**) Smaller unit having the exit point of the channel; (**c**) Larger unit having the 120° channel; (**d**) Experimental die setup along with its conductive heating arrangement; and (**e**) Hydraulic press machinery for ECAP accomplishment [13].

**Figure 5 materials-16-05158-f005:**
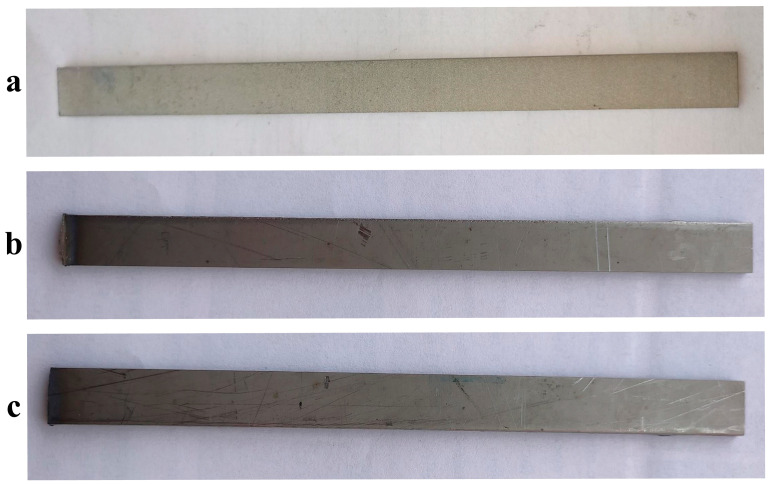
Magnesium RZ5 alloy sheet samples (**a**) before ECAP, (**b**) after ECAP two passes, and (**c**) after ECAP four passes.

**Figure 6 materials-16-05158-f006:**
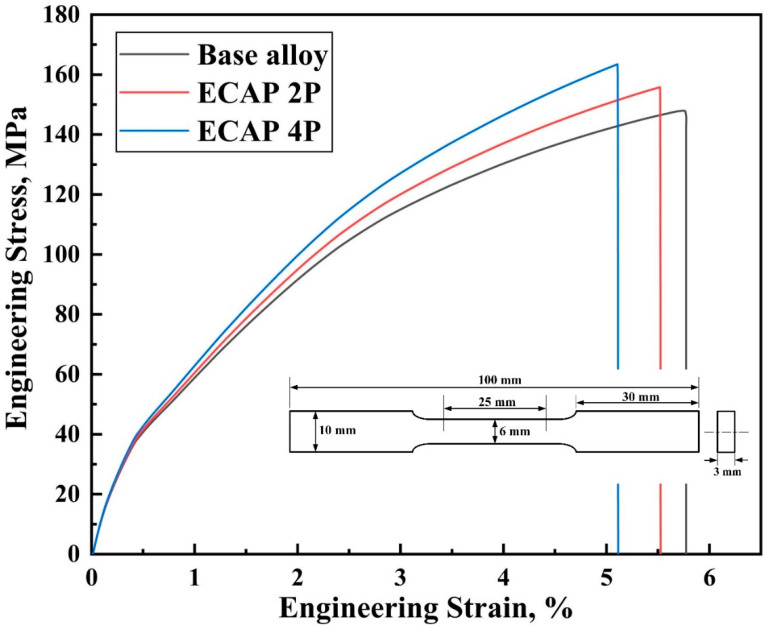
Tensile stress–strain curves of magnesium alloy sheet before and after the multi-pass ECAP technique.

**Figure 7 materials-16-05158-f007:**
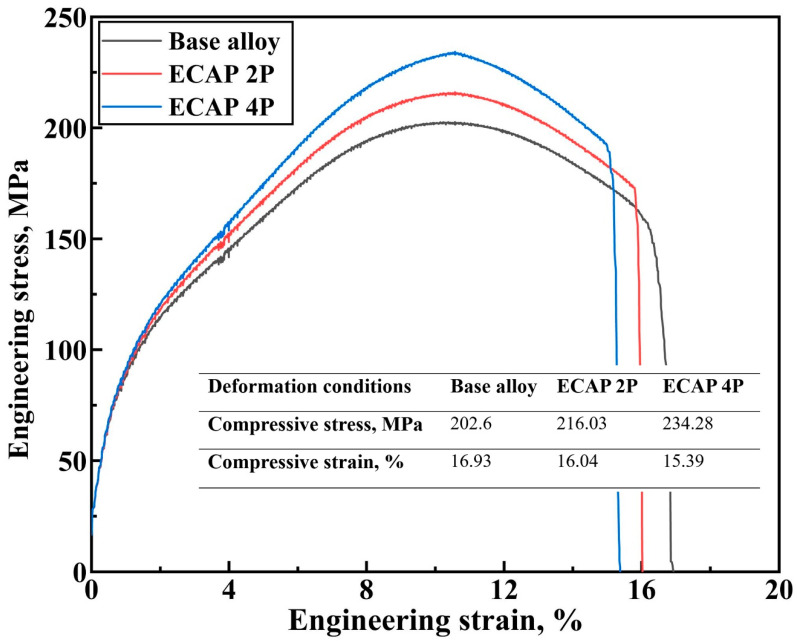
Room temperature compression testing of Mg-RZ5 alloy samples before and after the hot ECAP execution.

**Figure 8 materials-16-05158-f008:**
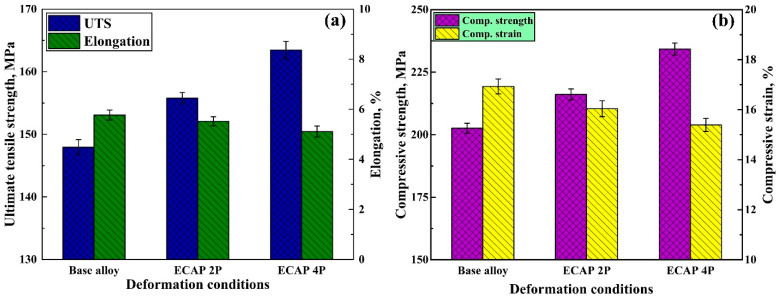
(**a**) Ultimate tensile strength and elongation tests; (**b**) compressive strength and strain testing of the magnesium alloy sheets, before and after the multi-pass ECAP.

**Figure 9 materials-16-05158-f009:**
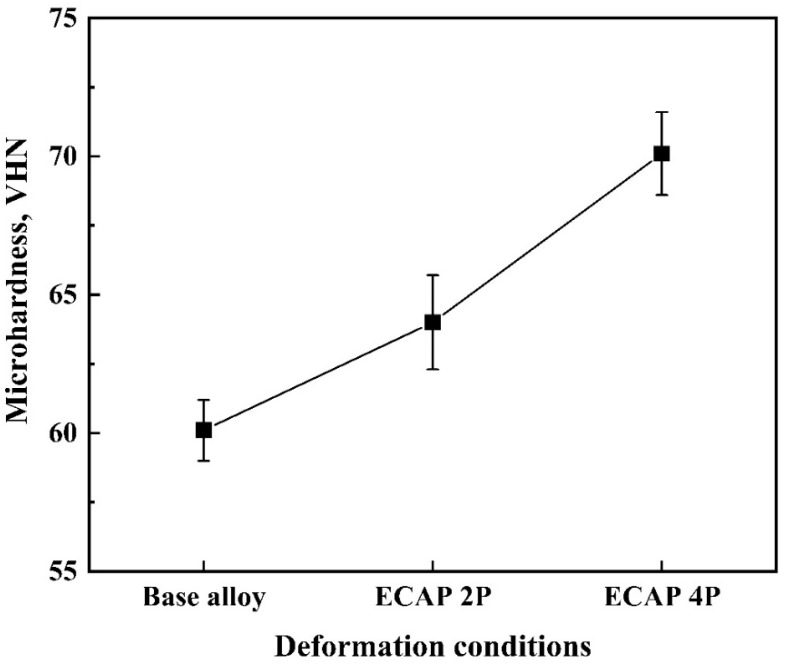
Microhardness tests of the Mg-RZ5 alloy sheets after multi-passes of ECAP.

**Figure 10 materials-16-05158-f010:**
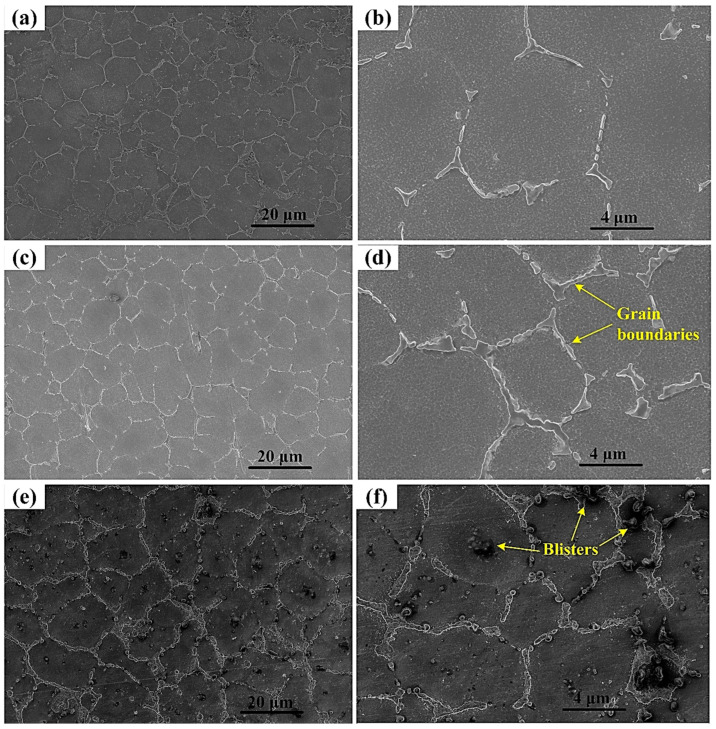
FESEM micrographs of (**a**,**b**) the base Mg-RZ5 alloy sheets and the corresponding images in their extruded direction (**c**,**d**) after ECAP second pass and (**e**,**f**) fourth pass.

**Figure 11 materials-16-05158-f011:**
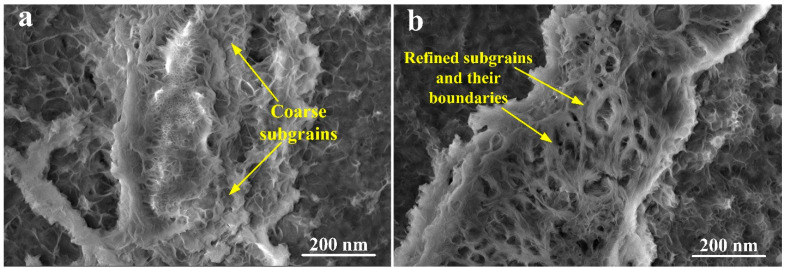
Emphasized FESEM images of grain boundaries in the magnesium RZ5 alloy (**a**) after second pass; and (**b**) after fourth pass of the hot ECAP technique.

**Figure 12 materials-16-05158-f012:**
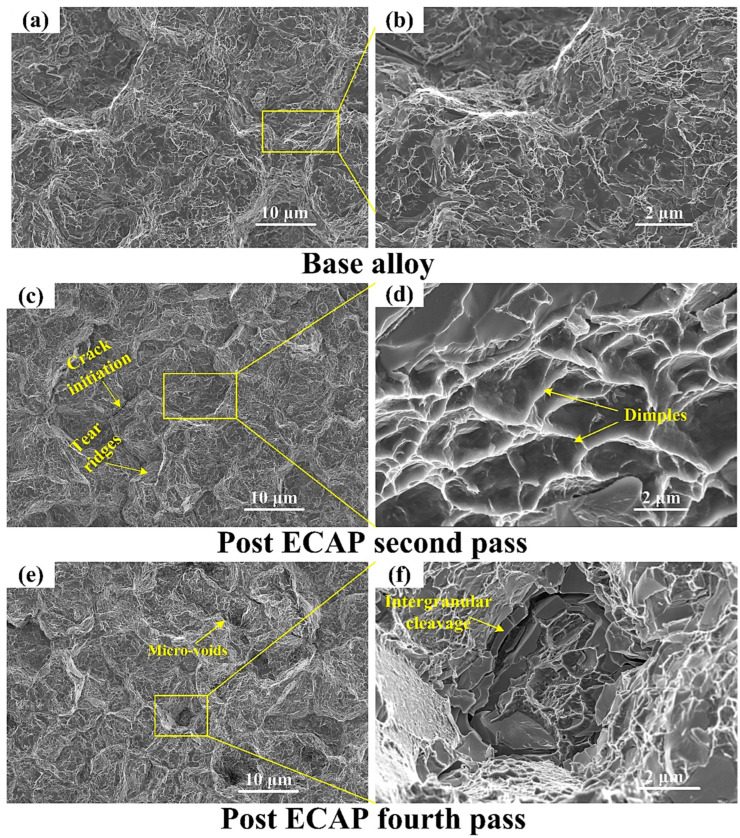
FESEM images of the fractured surfaces after the tensile testing of ECAPed Mg-RZ5 alloy sheet samples (**a**,**b**) for base alloy, (**c**,**d**) after two passes of ECAP, (**e**,**f**) after four passes of ECAP.

**Figure 13 materials-16-05158-f013:**
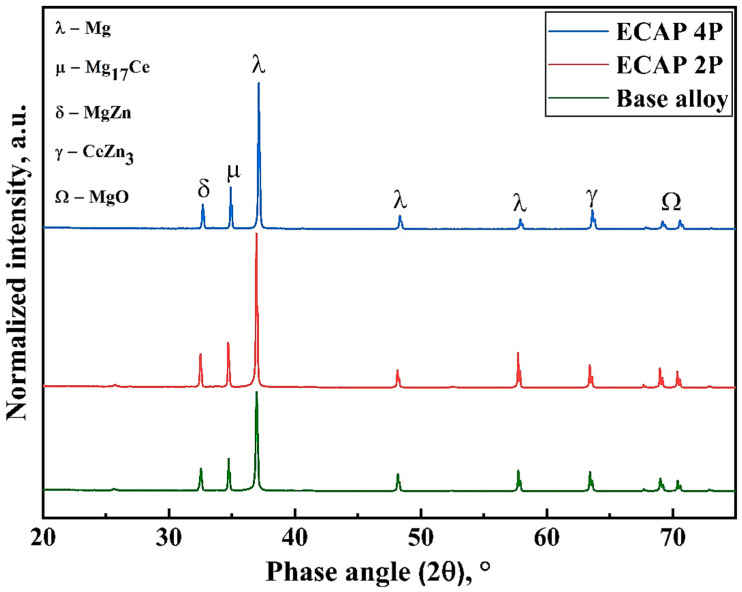
Superimposed XRD data for Mg-RZ5 alloy after two and four ECAP passes in accordance with its pre-ECAP form.

**Figure 14 materials-16-05158-f014:**
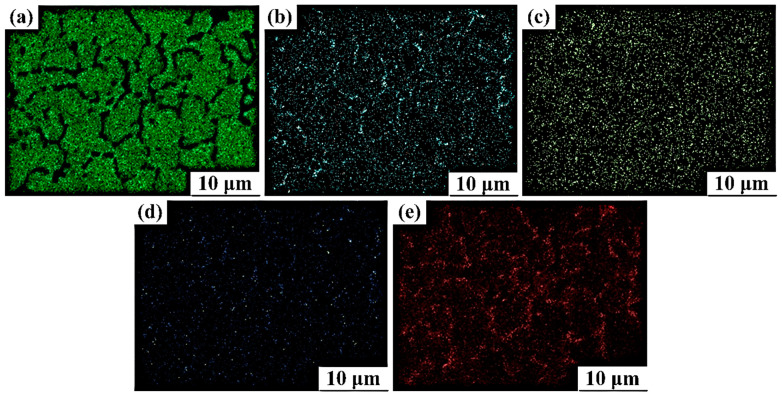
EDS images of element distribution in magnesium alloy after two passes of ECAP: (**a**) Mg; (**b**) Zn; (**c**) Zr; (**d**) Ce; and (**e**) O.

**Figure 15 materials-16-05158-f015:**
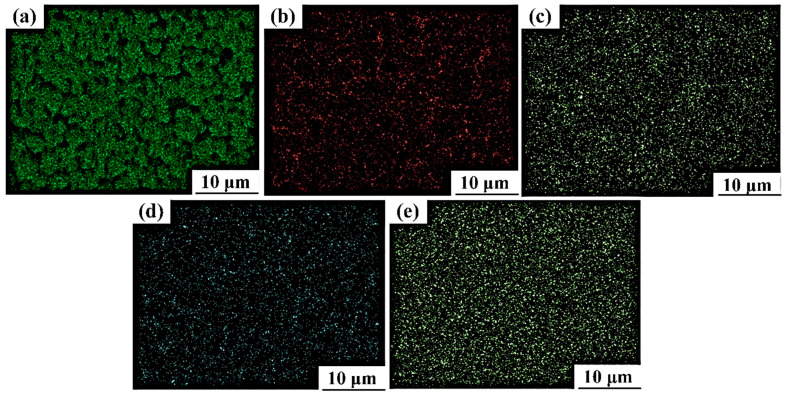
EDS images of element distribution in magnesium alloy after four passes of ECAP (**a**) Mg; (**b**) Zn; (**c**) Zr; (**d**) Ce; and (**e**) O.

**Figure 16 materials-16-05158-f016:**
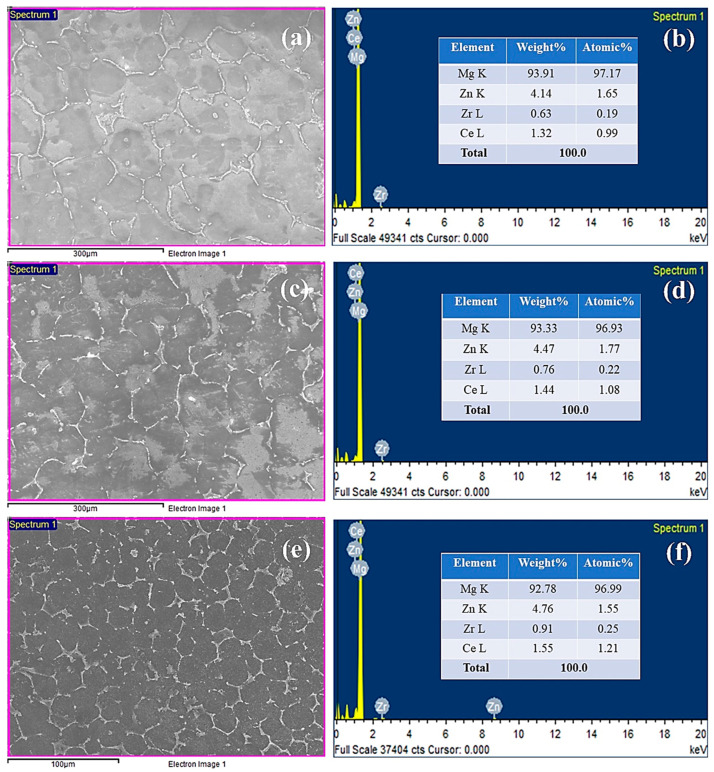
Point distribution spectra of magnesium alloy (**a**,**b**) at its base form (**c**,**d**) after ECAP 2nd pass (**e**,**f**) after 4th pass of ECAP.

**Table 1 materials-16-05158-t001:** Elemental composition of the magnesium RZ5 alloy.

Element	Zn	Zr	Rare Earth (Ce)	Impurities (Mn, Fe, Si, Cu)	Mg
**Weight (%)**	4.14	0.63	1.32	<0.005	Remaining

**Table 2 materials-16-05158-t002:** Average grain sizes of the Mg-RZ5 alloy with respect to its various deformation conditions.

Deformation Conditions	Average Grain Sizes (μm)	Hardness, VHN
Base form	164 ± 9.7	60.1 ± 1.1
ECAP 2P	51 ± 4.1	64 ± 1.7
ECAP 4P	12 ± 2.9	70.1 ± 1.5

## Data Availability

The raw/processed data required to reproduce these findings cannot be shared at this time due to legal or ethical reasons.

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
