# Peer review of "Investigation of Severe Plastic Deformation Effects on Magnesium RZ5 Alloy Sheets Using a Modified Multi-Pass Equal Channel Angular Pressing (ECAP) Technique"

_materials, 2023, doi:10.3390/ma16145158_

Round 1
Reviewer 1 Report
This manuscript analyzes the effect of the C route multiple pass hot ECAP technique on the microstructural evolution and room temperature mechanical properties of Mg-RZ5 alloy. The manuscript is well presented. However, this work needs to be further modified according to the comments and suggestion as below.
1. Please indicate where the 10° corner angle is.
2. What is the idea and processing of route C provided in Figure 2, please explain.
3. What do Figures 5 and 6 illustrate, please explain.
4. Please further improve the English writing throughout the article.
5. Multi-pass ECAP technique is presented in this paper. While, recent significant progresses about it are lost. For example, Experimental investigation on high-shear and low-pressure grinding process for Inconel718 superalloy[J]. The International Journal of Advanced Manufacturing Technology, 2020, 107(7). A novel high-shear and low-pressure grinding method using specially developed abrasive tools[J]. Proceedings of the Institution of Mechanical Engineers, Part B: Journal of Engineering Manufacture, 2021, 235(1-2).
Moderate editing of English language required
Reviewer 2 Report
The authors presented an article «Investigation of severe plastic deformation effects on magnesium RZ5 alloy sheets using a modified multi-pass ECAP technique». The manuscript is clear, relevant to the field and presented in a well-structured manner. The authors are advised to consider the following comments for this paper.
· Background and Introduction
Seemingly, a comprehensive literature review was given. However, they were just summarized one- by-one. The authors have to stop after writing each example and think about the contributions and lack of knowledge for each paper. After that, in the final lines of the introduction give the blank spots of the topic. Then it will be clear what did authors make differently from the open literature. More references should be included certainly the reference. Extreme plastic deformation, especially the studies using the ECAP technique, is a very popular deformation technique in the recent period. For this reason, if the authors increase the references section, it will not only enrich the discussion and literature of the study, but will also benefit the readers. For this reason, the following references are thought to be helpful for the authors. (1) An Overview of Deformation Path Shapes on Equal Channel Angular Pressing, (2) Influence of extrusion parameters on drilling machinability of AZ31 magnesium alloy.
In the last paragraph of the introduction section; What is the scientific novelty of the paper? What is the practical value? What makes this approach different from other researchers? Please specify. Gap and significance of the work must be included.
· 2. Materials and Methods
For this study in which die figures are given in detail, it would be more informative for the reader to add visual photographs in order to observe the deformation processes occurring in the samples after EKAP process, to observe the changes in their length and geometric changes.
“For conducting the present work, Mg-RZ5 alloy has been procured in the form of ingots from Hindustan Aeronautics Limited, Bangalore, India. Further, the ingots are machined and cut into sheets of 3 mm thickness precisely with the help of wire-cut electrical discharge machine.” Mg-RZ5 cylinder material cut from ingots. Shouldn't a normalisation anneal be applied before ECAP treatment? Does Mg-RZ5 have a processing history provided by the manufacturer?
In the study, why Mg-RZ5 was chosen, its usage areas, advantages and the contribution of the bushing to the use of this material should be explained.
Routes (A, BA, BC, C) are forgotten on the Figure 2.
What are the standards used in the experiments?
· 3. Results and Discussion
The results obtained should be explained by supporting the literature.
Similar conditions are also valid for Figure 9. Even in the FESEM micrographs given in Figure 9, it is seen that the grain structures grow after 2 passes.
· When the Figure c and e given in Figure 15 are examined, it is seen that the grain morphologies after ECAP should be orientated due to rotation. However, it is noteworthy that the grains are spherical in these microstructures.
· Conclusions
The conclusions need to be improved. The results are written long. It is necessary to more clearly show the novelty of the article. Add qualitative and quantitative results of your work. What is the difference from previous work in this area? Show practical relevance. What are the differences from previous works?
· Authors should carefully study the comments and make improvements to the article step by step. All changes should be highlighted in color.
Reviewer 3 Report
The manuscript "Investigation of severe plastic deformation effects on magnesium RZ5 alloy sheets using a modified multi-pass ECAP technique" has good potential in scientific terms and high practical significance. However, the following remarks to the manuscript:
- Lines 93-97: add 2-3 references to the literature;
- Line 121: indicate the device (company and country of manufacture) for energy dispersive x-ray spectroscopy;
- References to Tables 1 and 2 must be given before the tables themselves;
- Indicate the manufacturer and country of manufacture for the devices used.
